# Molecular and Cellular Mediators of Renal Fibrosis in Lupus Nephritis

**DOI:** 10.3390/ijms26062621

**Published:** 2025-03-14

**Authors:** Akshara Ramasamy, Chandra Mohan

**Affiliations:** Biomedical Engineering Department, University of Houston, 3517 Cullen Blvd, Room 2027, Houston, TX 77204, USA; akshararamasamy@utexas.edu

**Keywords:** systemic lupus erythematosus, lupus nephritis, renal fibrosis

## Abstract

Lupus nephritis (LN), a significant complication of systemic lupus erythematosus (SLE), represents a challenging manifestation of the disease. One of the prominent pathophysiologic mechanisms targeting the renal parenchyma is fibrosis, a terminal process resulting in irreversible tissue damage that eventually leads to a decline in renal function and/or end-stage kidney disease (ESKD). Both glomerulosclerosis and interstitial fibrosis emerge as reliable prognostic indicators of renal outcomes. This article reviews the hallmarks of renal fibrosis in lupus nephritis, including the known and putative drivers of fibrogenesis. A better understanding of the cellular and molecular processes driving fibrosis in LN may help inform the development of therapeutic strategies for this disease, as well as the identification of individuals at higher risk of developing ESKD.

## 1. Introduction

Systemic lupus erythematosus (SLE) is a chronic, multisystemic autoimmune disease with clinical symptoms that range from mucocutaneous manifestations to severe organ complications, most notably lupus nephritis (LN) [1]. Renal fibrosis continues to be a major cause of morbidity and mortality in LN, with 10% of patients developing end-stage kidney disease (ESKD) and needing either dialysis or kidney transplantation [1]. Despite the currently available treatment regimens, the risk of ESKD in proliferative LN has been as high as 44% over the past 15 years [2].

The current classification of LN pathology, based on International Society of Nephrology/Renal Pathology Society (ISN/RPS) criteria, is predominantly based on glomerular lesions [3]. However, the assessment of the renal pathology Activity Index (AI) and Chronicity Index (CI) adequately captures both glomerular and tubulo-interstitial lesions [4]. Using perhaps the largest cohort with the longest follow-up to date, Moroni et al. recently reported that the strongest histological predictors of poor long-term outcomes in LN are glomerulosclerosis (OR = 3.1), fibrous crescents (OR = 6.8), and tubular atrophy/interstitial fibrosis (OR = 2.4–3.2), all of which are manifestations of renal fibrosis, currently best captured using the CI [5]. In contrast, baseline serum creatine exhibited an OR of 1.7 in this same cohort [5]. Thus, there is an urgent need to better understand the molecular mechanisms leading to fibrotic changes in LN, both in glomeruli and in tubulo-interstitial regions.

While significant advances have been made in our efforts to effectively manage LN, several barriers remain, including knowledge gaps in our understanding of the mechanisms leading to fibrosis in LN. As no targeted treatments currently exist to slow renal fibrosis, it remains a challenge to translate the highly limited research surrounding renal fibrogenesis into promising antifibrotic therapies for clinical use. Here, we discuss the molecular signatures of renal fibrosis, as well as known and potential molecular contributors to fibrogenesis, with a focus on LN.

## 2. Pro-Fibrotic Markers of Renal Fibrosis

Several biomarkers have been implicated in LN-associated fibrosis, including α-SMA, collagen, fibronectin, fibroblast-specific protein-1 (FSP-1), and vimentin. While these markers have been extensively studied in other chronic kidney diseases (CKDs), their role and expression in LN fibrosis remain less well characterized. Below, we summarize the expression profiles of these markers in LN (Table 1).

### 2.1. α-Smooth Muscle Actin (α-SMA)

α-SMA is a well-established marker of activated myofibroblasts, which play a central role in fibrosis. In healthy kidneys, α-SMA is present in interstitial peritubular regions but is absent from glomeruli [6]. However, in proliferative glomerulonephritis and all forms of glomerulonephritis studied, α-SMA staining is observed in the mesangial and interstitial compartments, indicating its role in fibrotic remodeling [6].

Similarly, in murine lupus models, α-SMA expression is elevated in glomeruli (in the pristane-induced lupus model), while in MRL/lpr mice, α-SMA is observed in proximal tubules and vascular walls, as summarized in Table 1 [7,8]. Studies involving LN patients have shown increased renal α-SMA expression, with α-SMA+ spindle-shaped fibrocytes documented in peripheral blood and renal tissue [9,10]. Notably, the number of renal fibrocytes expressing α-SMA correlates significantly with interstitial fibrosis severity and renal dysfunction [10]. Importantly, in Class IV LN patients, mesangial α-SMA expression increases, but interstitial α-SMA expression correlates better with chronicity indices and predicts renal outcomes [11].

Despite its strong association with fibrosis, α-SMA lacks specificity, as it is not restricted to fibroblasts but is also expressed by vascular smooth muscle cells, pericytes, and other mesenchymal-derived cells [12]. This lack of exclusivity raises concerns about its ability to differentiate LN fibrosis from other pathological processes, such as vascular remodeling. Additionally, while α-SMA is a widely accepted fibrosis marker in CKD, such as IgA nephropathy (IgAN), diabetic nephropathy (DN), and proliferative/non-proliferative glomerulonephritis, its cellular origins and pathogenic relevance in LN fibrosis remain to be fully established [13].

### 2.2. Collagen

The composition of the tubular renal interstitium consists of collagens I, II, III, V, VI, VII, and XV, with type IV collagen being the most prevalent protein in the basement membrane and mesangial matrix of healthy adult kidneys [14].

In lupus-prone murine models, collagen expression undergoes dynamic changes throughout disease progression. Among mice with chronic graft-versus-host disease, a model that mimics LN, increased mesangial expression of collagen IV α1 and α2 was found during the early stages of the disease, but only late in the glomerular basement membrane [15]. However, the expression of collagen IV α4 increased in the glomerular basement membrane throughout the course of the disease, but not in the mesangial matrix [15]. In lupus-prone B/W mice, collagen IV α1 and α2 were highly upregulated at the proteinuric stage of LN, while collagen IV α6 was increased throughout disease progression compared to healthy mice [16].

Similarly, interstitial collagen type III (PRO-C3) and type VI (PRO-C6) have been reported to be significantly elevated in the serum and urine of LN patients [17]. Within a subgroup of LN patients with preserved glomerular filtration rate (GFR), high collagen I (COL1) mRNA expression emerged as an independent predictor of subsequent renal outcomes [18]. Strikingly, LN patients exhibiting increased COL1 gene expression displayed a greater than four-fold increased risk of developing progressive renal failure [18]. Renal biopsies of cortical tissue from LN patients revealed that a high collagen matrix index was associated with both relapse and progression to ESKD, although the specific cells expressing collagen were not studied [19]. More specifically, the fibrillary collagen index emerged as a predictive tool for disease progression, as quantified by the doubling of serum creatinine, and relapse [19].

Extensive studies conducted involving patients with DN or IgAN, renal transplant recipients, and those with other CKDs have shown that collagen expression correlates with disease progression and demonstrates potential as a biomarker [20]. Although collagen accumulation is a hallmark of renal fibrosis, its specific expression patterns in LN versus other CKDs remain unclear. The cited studies suggest that collagen accumulation may predict renal disease progression and relapse in LN, but further research is needed to determine whether specific collagen isoforms have differential predictive potential in LN fibrosis.

### 2.3. Fibroblast-Specific Protein-1 (FSP-1)

Fibroblast-specific protein-1 (FSP-1), also known as S100A4, was among the first proteins studied in the context of renal fibrosis. Strutz et al. first reported that FSP-1 is expressed in fibroblasts but not in tubular epithelia [21]. Interestingly, in healthy kidneys, FSP-1+ cells are scarce; however, in a murine model of renal fibrosis induced by unilateral ureteral obstruction, FSP-1+ fibroblasts were abundantly found in the interstitium and tubular epithelia [22].

Elevated urine FSP-1 has been reported as a diagnostic biomarker of LN [23]. All assessed urine S100 protein levels declined as LN improved, with FSP-1/S100A4 showing the greatest decrease [23]. However, FSP-1 is not specific to fibroblasts, with staining reported in mononuclear cells, podocytes, and distal tubular epithelial cells within LN kidneys [23]. Additionally, FSP-1 expression was shown to be higher in podocytes of glomeruli with cellular crescents from individuals with lupus nephropathy in another study [24]. As opposed to renal and urinary FSP-1, FSP-1 is not significantly elevated in the peripheral blood cells of SLE patients [25].

FSP-1 has been shown to enhance fibrosis via TGF-β/Smad3 signaling, a pathway implicated in other CKDs, including IgAN, DN, and focal segmental glomerulosclerosis (FSGS) [26]. Similar mechanistic studies are lacking in LN, making this an important area for further investigation.

### 2.4. Fibronectin

In renal fibrogenesis, fibronectin is one of the first extracellular matrix (ECM) proteins to be deposited, co-localizing with collagen and activating integrins [27]. In healthy kidneys, the glomerular mesangium, Bowman’s capsule, and peripheral capillary loop exhibit low levels of fibronectin; however, this expression markedly increases during pathology [28]. Importantly, fibronectin activates fibroblast proliferation [29]. Fibronectin is highly implicated in fibronectin glomerulopathy, IgA nephropathy (IgAN), and other CKDs [20].

Increased tubulointerstitial fibronectin in LN patients has been demonstrated to co-localize with IgG deposition [30]. Increased glomerular expression of fibronectin has also been observed in active murine LN, with Yung and colleagues corroborating this finding in LN patients, where fibronectin expression was elevated and co-localized with IgG deposition [31,32]. Together, these results indicate a link between immune complex deposition and fibrotic remodeling. However, fibronectin’s diagnostic utility in LN fibrosis remains uncertain, as some studies have failed to detect elevated fibronectin in LN glomeruli [33].

### 2.5. Vimentin

In healthy adult kidneys, vimentin is expressed in the mesangial and epithelial cells of the glomeruli but not in the tubulointerstitium [6]. In addition, fibroblasts are activated by vimentin, which is only expressed de novo in activated but not dormant fibroblasts of adult kidneys [34].
ijms-26-02621-t001_Table 1Table 1Elevated markers of LN fibrosis in different cell types.
ReferenceMesangiumPodocyteGlomerulusTubuleInterstitiumBloodmRNAUrineMacrophages**Murine Model**









Pristane Induced BALB/c[7]A







MRL/lpr[8]


A
A


GvHD[15]C
C





NZBxNZW F1 (B/W)[16]

C





NZBWF1/J[30]F







**Human LN**[10]



A,CA


[11]A


A



[17]




C
C
[18]





C

[23]
S
S



S[24]
S






[30]


F,CF



[32]F



F


[35]






V
Abbreviations: A: α-SMA, C: Collagen, F: Fibronectin, S: FSP-1, V: Vimentin, LN: Lupus Nephritis.

In LN patients, elevated urine vimentin has been found, suggesting its role as a viable biomarker of the disease [35]. Interestingly, vimentin has emerged as a target for the in situ immune response. High titers of anti-vimentin antibodies are associated with severe tubulointerstitial inflammation, raising the possibility that vimentin is not only a fibrosis marker but also involved in LN pathogenesis [36]. Others have reported similar findings, validating the likelihood of vimentin as a pathogenic autoantigen in the lupus nephritic milieu [37,38].

In patients with LN and other CKDs, such as IgAN, minimal change disease, hypertensive nephropathy, and DN, elevated urinary vimentin mRNA levels correlate with increased proteinuria, renal fibrosis scores, and a decrease in GFR, highlighting its potential as a diagnostic marker for renal outcomes [39]. However, studies detailing the spatial distribution of vimentin in LN kidneys and its cellular origins are limited, necessitating further study.

### 2.6. Comparative Analysis of Fibrosis Markers in LN

Among the fibrosis markers examined, α-SMA and collagen are the most consistently upregulated in LN fibrosis across multiple studies. Fibronectin, while an early marker of fibrogenesis, has shown inconsistent results in LN studies; however, not many LN studies have examined this marker in the kidneys. FSP-1 and vimentin represent additional markers of renal fibrosis, but their expression profiles have not been systematically examined in LN. A major limitation is the lack of large-scale comparative proteomic studies assessing their respective spatial distribution and cellular origins in LN. The advent of spatial proteomic analyses of LN, as reported recently, will help address this knowledge gap [40]. Additionally, longitudinal studies are needed to determine which fibrosis markers best predict long-term renal outcomes in LN.

## 3. Triggers and Signaling Pathways Mediating Renal Fibrosis

### 3.1. TGF-β

Transforming growth factor-beta (TGF-β) is the most extensively studied and predominant profibrotic mediator in various chronic kidney disorders, including LN [41].

Among its isoforms, TGF-β1 is particularly recognized for driving renal fibrosis. Early studies, such as those by Yamamoto et al., demonstrated a correlation between TGF-β expression and disease severity in LN, with higher levels in diffuse proliferative LN compared to focal proliferative LN [42]. More recent studies confirm this association, showing that urinary TGF-β levels correlate significantly with fibrosis severity and the chronicity index [43]. Furthermore, LN patients had decreased serum TGF-β1 levels, while their urine TGF-β1 levels were significantly higher, suggesting that urinary TGF-β1 may be of renal origin [44]. Importantly, an unbiased proteomic screen of 1000 proteins in LN urine also identified TGF-β1 and TSP-1 (which converts latent TGF-β1 to its active form) as among the best markers of active LN [45]. Taken together, these findings demonstrate the potential of urinary TGF-β to serve as a robust biomarker of lupus nephritis.

Given its well-established role in renal fibrosis, targeting TGF-β signaling remains a major therapeutic focus. In the context of LN, the precise cell types within the kidneys that are the targets of this cytokine have yet to be systematically investigated.

### 3.2. Smad Signaling

The canonical TGF-β/Smad pathway plays a central role in mediating fibrotic responses, yet its specific contribution to LN pathogenesis remains incompletely understood.

Positive Smad3 and Smad7 immunostaining has been reported in LN kidneys [46]. Smad3 levels in urine exosomes are higher in LN patients compared to controls, correlating with the renal pathology chronicity index [47]. In all of the glomerular disorders investigated in a study by Schiffer et al., including LN, Smad7 was found to be up-regulated in podocytes, while Smad6 (an inhibitor of TGF-β signaling) was significantly decreased [48]. However, another study found Smad7 to be reduced in SLE patients compared with healthy controls [49]. Additional research is clearly necessary to fully elucidate the roles of activating and inhibitory Smad proteins in LN.

In other CKDs, such as DN, obstructive kidney disease, 5/6 nephrectomy, and hypertensive nephropathy, the Smad family has been heavily implicated in fibrogenesis. Their role in post-translational modifications and signaling pathways significantly correlates with disease outcomes, underscoring the importance of targeting Smad-dependent pathways therapeutically [50].

### 3.3. Wnt/β-Catenin Signaling

The Wnt signaling pathway is a cell-to-cell communication system that has been preserved throughout evolution. While relatively quiescent in healthy adult kidneys, the signaling of this pathway is stimulated during renal damage [51].

Aberrant Wnt/β-catenin signaling has been implicated in LN. While Orme and colleagues found decreased levels of DKK-1, an inhibitor of the WNT pathway, in lupus-prone mice, Wang et al. found DKK-1 to be elevated in the plasma of LN patients [52,53]. Furthermore, the gene expression of β-catenin significantly correlated with creatinine clearance and the renal pathology chronicity index [53]. Strikingly, β-catenin mRNA expression was higher in patients without renal interstitial fibrosis than in those with renal interstitial fibrosis [53]. Compared to TGF-β/Smad signaling, Wnt/β-catenin’s role in LN fibrosis is less well defined.

Upregulated Wnt/β-catenin signaling in other CKDs, such as autosomal dominant polycystic kidney disease, DN, and FSGS, has been more thoroughly investigated; however, studies have also demonstrated a protective role of Wnt signaling in acute renal failure and high glucose-mediated cell apoptosis, underscoring the nuanced and sometimes contradictory nature of Wnt signaling in fibrotic renal diseases compared to other conditions [54].

### 3.4. Emerging Signaling Pathways: mTOR and PPARγ

While TGF-β/Smad and Wnt/β-catenin represent the most extensively characterized pathways in renal fibrosis, recent studies highlight the roles of mechanistic target of rapamycin (mTOR) and peroxisome proliferator-activated receptor gamma (PPARγ). The mTOR pathway, particularly mTORC1/2, has been found to be significantly activated in LN kidneys, with mTORC2 activity closely correlating with fibrosis [55]. Meanwhile, PPARγ agonists show promise in reducing TGF-β1-induced glomerulosclerosis and tubulointerstitial damage, as well as suppressing inflammatory cells in murine models, suggesting a potential antifibrotic role [56,57].

### 3.5. Comparative Analysis of Signaling Pathways

Among the major signaling pathways implicated in LN fibrosis, TGF-β remains the predominant effector, primarily exerting its effects through Smad-dependent mechanisms. Wnt/β-catenin signaling, while clearly involved, appears to have a more nuanced and context-dependent role. The emerging contributions of mTOR and PPARγ further suggest that LN fibrosis results from a complex interplay of multiple pathways rather than a single dominant mechanism. Indeed, emerging evidence demonstrates that other signaling pathways, such as SHH, Notch, PI3K/AKT, JAK/STAT, NF-κB, and MAPK, are implicated in ECM buildup and renal fibrosis [58,59]. Figure 1 presents an overview of the signaling/pathogenic mechanisms leading to fibrosis in LN. However, each pathway has been examined in a compartmentalized fashion, with little effort to integrate these different pathways to delineate their interdependence. Future studies should focus on this network of underlying signaling pathways, aiming to identify hubs that may be amenable to therapeutic intervention.

## 4. Other Pathogenic Drivers of Renal Fibrosis

### 4.1. Growth Factors

Various growth factors collectively contribute to renal fibrosis by promoting fibroblast activation, extracellular matrix (ECM) deposition, and sustained pro-fibrotic signaling. Connective tissue growth factor (CTGF) acts as a central mediator by amplifying TGF-β signaling and interacting with ECM proteins, thereby reinforcing fibrotic responses [60].

Platelet-derived growth factor (PDGF) exacerbates fibrosis by stimulating mesangial cell proliferation and recruiting fibroblasts, leading to excessive matrix accumulation [60]. Fibroblast growth factor (FGF) family members, particularly FGF2, have been implicated in fibrosis because of their roles in promoting cell proliferation and differentiation in response to injury [60].

Vascular endothelial growth factor (VEGF), although essential for endothelial survival, can contribute to fibrosis when dysregulated, as seen in diabetic kidney disease [60]. Hepatocyte growth factor (HGF), despite its regenerative functions, may also play a dual role in fibrosis by modulating epithelial-to-mesenchymal transition (EMT) [60]. Similarly, epidermal growth factor (EGF) has been linked to tubular cell proliferation and repair; however, its overactivation can contribute to fibrotic scarring [60].

Finally, insulin-like growth factor-1 (IGF-1) influences renal fibrosis through its effects on cell survival, proliferation, and matrix synthesis [60]. Collectively, these growth factors and additional cytokines drive a pathological cycle of injury, inflammation, and fibrotic remodeling, highlighting their potential as therapeutic targets in renal disease [61,62]. Most of the abovementioned growth factors have been studied in the context of other CKDs, while studies on LN are scarce.

### 4.2. Cellular Stress

Additional contributors to renal fibrosis include hypoxia, altered lipid metabolism, and cell cycle arrest. Peritubular capillary rarefaction, observed in various kidney disorders, reduces oxygen supply to tubulointerstitial regions, leading to persistent hypoxia and dysregulated tissue repair, ultimately resulting in ECM accumulation [63,64]. Additionally, impaired fatty acid oxidation (FAO) and excessive intracellular lipid deposition within proximal tubular epithelial cells have been identified in both murine and human models of tubulointerstitial fibrosis [65]. Furthermore, kidney injury-induced cell cycle arrest can drive tubular epithelial cells (TEC) toward a pro-fibrotic secretory phenotype, contributing to disease progression [66].

Recent findings have clarified how these mechanisms intersect in LN fibrosis, with hypoxia playing a key role. Chen et al. demonstrated that extensive hypoxia in LN kidneys, particularly in the tubulointerstitial compartment, is strongly associated with increased expression of hypoxia-inducible factor-1α (HIF-1α) [67]. This transcription factor promotes metabolic reprogramming, shifting renal-infiltrating immune cells toward glycolysis and enhancing T cell survival and effector function, thereby worsening inflammation and fibrosis [67]. HIF-1α also upregulates profibrotic mediators such as TGF-β and osteopontin (OPN), reinforcing extracellular matrix deposition and myofibroblast activation [67].

Beyond immune cell function, metabolic adaptations driven by HIF-1α, including enhanced glycolysis and amino acid metabolism, further sustain inflammation. Notably, renal-infiltrating T cells in LN exhibit heightened proline metabolism via proline dehydrogenase (PRODH), a pathway that supports glycolysis by maintaining the NADH/NAD^+^ redox balance, ultimately fueling cytokine production and fibrosis [67].

Targeting metabolic and inflammatory pathways linked to fibrosis presents a promising therapeutic avenue. Inhibiting HIF-1α with PX-478 in murine LN models reduced T cell infiltration, reversed cortical hypoxia, and attenuated fibrosis, supporting the role of hypoxia-driven immune activation in disease progression [67]. Meanwhile, interventions aimed at restoring FAO or modulating cell cycle arrest may further mitigate fibrosis by addressing tubular cell dysfunction and lipid accumulation. Integrating these approaches could provide a more comprehensive strategy for halting fibrotic progression in LN.

### 4.3. MicroRNAs and Long Noncoding RNAs

Emerging evidence indicates that renal fibrosis pathways may also be impacted by microRNAs (miRs). TGF-β1 signaling may regulate microRNAs such as miR-21, miR-150, and miR-192, which function as downstream factors promoting excessive collagen deposition and tissue fibrosis [68,69,70]. Conversely, some microRNAs, such as miR-29, may play a protective role in the pathogenesis of renal fibrosis by acting as antifibrotic factors [71,72]. Long non-coding RNAs (lncRNAs), which are RNA molecules longer than 200 nucleotides that do not encode proteins, have emerged as key regulators of gene expression in various diseases, including LN and fibrosis.

Recent studies highlight the significant role of lncRNAs in modulating immune responses and renal injury. For instance, Zhou et al. identified Smad3-dependent lncRNAs that are linked to renal inflammation and fibrosis, suggesting a functional relationship between these lncRNAs and progressive kidney damage [73]. In the context of LN, lncRNAs have been implicated in disease progression. Findings indicate that urinary lncRNAs such as MEG3 and ANRIL inversely correlate with proteinuria severity, while MALAT1 is associated with histological activity [74]. Together, these insights underscore the role of noncoding RNAs in LN and call for more comprehensive studies examining the role of non-coding RNAs in renal fibrosis initiation and progression.

### 4.4. Mitochondrial Dysfunction

Mitochondrial dysfunction is a pivotal driver of renal fibrosis, particularly in proximal TECs, which are highly dependent on mitochondrial oxidative phosphorylation. Under fibrotic conditions, metabolic reprogramming occurs, exemplified by the downregulation of SIRT3, a key mitochondrial deacetylase [75]. The subsequent hyperacetylation of mitochondrial proteins impairs essential metabolic pathways and contributes to TEC dysfunction and fibrosis progression [75].

Mitochondrial gene regulation further compounds this dysfunction. In LN, a specific mitochondrial gene, MTND5, exhibits altered expression patterns, particularly through the downregulation of its circular RNA form [76]. Circular RNAs, by modulating microRNAs, are crucial regulators of gene expression. Under normal conditions, circular MTND5 downregulates microRNA6812, which, when persistently upregulated, activates genes associated with fibrosis. The downregulation of circular MTND5 in LN may contribute to an exacerbated fibrotic response by allowing microRNA6812 to remain active, further promoting the expression of fibrosis-related genes [76]. Together, these mitochondrial disturbances—both metabolic and regulatory—underscore the complex interplay contributing to renal fibrosis, highlighting the potential of targeting mitochondrial pathways to mitigate fibrotic progression.

### 4.5. Race, Ethnicity, and Genetic Factors

The severity and progression of LN to renal fibrosis vary across racial and ethnic groups. LN is more prevalent among African American, East Asian, and Hispanic individuals with SLE than among those of European descent [77]. Additionally, disease progression differs, with Black patients being more likely to develop tubulointerstitial fibrosis and tubular atrophy [78]. Genetic associations also vary by ethnicity, with distinct LN-related variants identified in different populations. For instance, the single nucleotide polymorphism (SNP) rs8091180 in NFATC1 showed the strongest association in African Americans, while TRIM family genes were linked to LN in South Europeans, and TTC34 was implicated in Hispanics [77]. Pathway analysis further highlighted ethnic-specific therapeutic targets, such as mTOR signaling in Asians and SMAD/TGF signaling in North Europeans, both of which play key roles in LN fibrosis [77]. Collectively, these findings emphasize the importance of factoring in ethnic/genetic factors in studying the mechanisms underlying LN fibrosis and in formulating personalized treatments.

## 5. Cellular Mediators of LN Fibrosis

### 5.1. Cellular Mediators of Tubulointerstitial Fibrosis

#### 5.1.1. Activated Resident Fibroblasts

Fibroblasts are stellate-shaped cells with collagen-containing granules and actin filaments [79]. In healthy kidneys, fibroblasts are found in the interstitial space between the capillaries and tubules [34]. Fibroblasts serve as a source of erythropoietin (EPO) and synthesize a basal level of ECM components to maintain the homeostasis of the interstitial matrix [79]. However, the source of myofibroblasts in renal fibrosis has been a topic of ongoing debate and remains unresolved. Evidence suggests that these cells may originate from various progenitors, including bone marrow-derived fibroblasts, TECs, endothelial cells, pericytes, and interstitial resident fibroblasts [80]. In other forms of CKD, such as chronic allograft rejection, increased transdifferentiation of resident fibroblasts into contractile myofibroblasts has been observed, driven by TGF-β and other growth factors [81].

Renal fibroblasts in LN have received little attention. Recently, in a study by Kim et al., the peripheral blood of LN patients with interstitial fibrosis exhibited a higher abundance of spindle-shaped fibrocytes [10]. Indeed, in the tubulointerstitium of patients with LN, renal fibrocytes expressing collagen and α-SMA were identified, and their numbers were substantially correlated with interstitial fibrosis and renal failure [10]. Stimuli reported to activate renal fibroblasts, including IL-1 family cytokines, CTGF, TGF-β, and PDGF, and/or their receptors, are significantly elevated in LN patients [82,83,84]. Among these, serum CTGF was also noted to be positively correlated with interstitial fibrosis in LN [84]. Extrapolating from other studies, increased activated renal fibroblasts are likely to be important in LN, though their origins and impact have not been systematically investigated.

#### 5.1.2. Renal Tubular Epithelial Cells

Renal TECs play multifaceted roles, possessing innate immune properties, and responding to various insults by producing bioactive mediators that drive interstitial inflammation and fibrosis. These cells, recognized as key players in inflammatory and fibrogenic processes in other renal disorders, are considered the link between acute kidney injury and CKD due to their maladaptive repair mechanisms, including phenotypic changes that lead to sustained inflammation, fibroblast activation, and ECM deposition [66].

In LN kidneys, TECs exhibit a secretory phenotype, actively producing soluble factors like pro-inflammatory cytokines and IFN-1 [85]. Anti-dsDNA antibodies have also been reported to induce pro-fibrotic changes in proximal TECs in LN, including increased expression of TGF-β, fibronectin, and collagen [30].

Wang et al. linked the presence of tubular basement membrane (TBM) immune complex (IC) deposits with more active disease in LN, including higher serum creatinine levels and poorer prognosis [86]. The deposition of ICs in the TBM can activate complement, which has been correlated with more severe tubulointerstitial pathology, including interstitial fibrosis and tubular atrophy [87]. Interestingly, the antibody subclass composition of TBM IC deposits in LN kidneys differed between TBM and glomerular deposits, indicating the independent formation of tubular ICs [88]. Investigating the cellular and molecular pathways triggered by these IC deposits is a growing area of research. Further studies are clearly warranted to comprehensively study the cell types contributing to tubulo-interstitial fibrosis in LN.

## 6. Macrophages

Macrophages exhibit a bifaceted role in immune responses: M1 cells initiate pro-inflammatory reactions, while M2 cells are essential for resolving inflammation and facilitating tissue repair. Thus, maintaining macrophage plasticity is crucial for sustaining homeostasis, as an imbalance in this homeostasis can lead to fibrosis in LN.

Recent investigations have implicated M2 macrophages in LN fibrosis. Olmes et al. demonstrated a correlation between the prevalence of M2a-like (CD206+/CD68+) and M2c-like (CD163+/CD68+) macrophages with tubular injury, glomerulosclerosis, crescent formation, and interstitial fibrosis/tubular atrophy, suggesting a contributory role in fibrogenesis [89]. Supporting these findings, Zhang and colleagues, in a study of 228 SLE patients, found significantly elevated levels of urine CD163, presumably sourced from M2 macrophages, in active LN compared to healthy controls. This increase was significantly correlated with cellular crescents and interstitial inflammation in patient biopsies, with CD163 effectively distinguishing active LN from other forms of SLE [90]. In accordance with the abovementioned findings, subsets of M2 macrophages, identified by CD163+ and CD206+ markers, were predominantly localized within fibrous crescents and were notably more abundant in LN and ANCA-associated vasculitis compared to other renal disorders studied [91,92].

### Regulating Macrophage Polarization in LN Fibrosis

While M2 cells are classically involved in tissue repair and anti-inflammatory processes, their prolonged functioning in CKD, including LN, can disrupt renal homeostasis and contribute to fibrosis. The relative contribution of macrophages to glomerular and tubulointerstitial fibrosis in LN, compared to that of resident renal cells, remains ambiguous. Additionally, the main molecular triggers responsible for M1/M2 skewing in vivo and the essential molecules employed by these macrophages to inflict renal damage in LN are still under-explored.

Recent studies have emphasized the role of chemokine networks, particularly CCL2, CSF-1, and fractalkine (CX3CL1), in recruiting monocytes to inflamed kidneys, where microenvironmental factors dictate their differentiation into pro-inflammatory M1-like or regulatory M2-like macrophages [93]. Persistent immune activation skews macrophages toward an M2 phenotype, in which the predominant M2a (CD206+) and M2c (CD163+) subsets secrete profibrotic mediators such as TGF-β, PDGF, and OPN [93].

Beyond immune-mediated polarization, metabolic reprogramming has emerged as another determinant of macrophage function in LN fibrosis. Profibrotic M2 macrophages in LN kidneys exhibit increased oxidative metabolism and lipid accumulation, which further sustain their activation and cytokine production [93]. This metabolic shift is influenced by signaling pathways such as PPARγ and HIF-1α, both of which have been previously implicated in driving LN fibrosis [93].

In experimental models of CKD, such as UUO and DN, macrophages exhibit a spectrum of activation states, with M1 macrophages driving early-stage inflammation and M2 macrophages becoming more prominent in later stages, contributing to fibrosis and glomerulosclerosis. However, persistent inflammation often limits the differentiation and function of M2 macrophages, leaving the mechanisms governing their role in tissue repair and fibrosis unclear [94].

## 7. Lymphocytes

Other than their antibody secretion role, B cells may contribute to renal fibrosis by secreting pro-inflammatory cytokines such as IL-6 and TNF-α, which stimulate fibroblast activation and extracellular matrix deposition, as well as by elaborating chemokine ligand-2 (CCL2), which promotes monocyte and macrophage infiltration into the kidney [95,96]. In experimental models of renal fibrosis, depletion of B cells using anti-CD20 monoclonal antibodies significantly reduced inflammation, monocyte recruitment, and collagen deposition, indicating a direct contribution of B cells to fibrosis progression [96]. Moreover, the formation of tertiary lymphoid structures (which harbor B cells) within the kidneys further sustains chronic immune activation, promoting fibrosis even in the absence of acute inflammation [95]. These findings have significant implications in LN, where B cells are already known to contribute to disease progression through autoantibody and immune complex production. Interventions targeting B cell-derived cytokines or chemokines may offer new therapeutic avenues to dampen fibrosis in LN. While LN patients respond well to B cell-depleting therapies, the role of such interventions in directly mitigating LN fibrosis has not been well studied [97]. Understanding potential antibody-independent mechanisms by which B cells may drive fibrosis is an important area for investigation.

Growing research has clarified the role of T cells in LN fibrosis. CD8+ T cell infiltration into the tubulointerstitial compartment is strongly correlated with markers of disease severity, including increased serum creatinine, proteinuria, and glomerulosclerosis [98]. High levels of tubulointerstitial CD8+ T cells have been identified as independent predictors of ESKD progression, highlighting their contribution to renal deterioration [98]. Moreover, CD4+ T cells, particularly Th1 cells, exacerbate LN through the production of pro-inflammatory cytokines such as IFN-γ and IL-12, which further amplify immune activation and renal injury [99]. Targeting T cell infiltration and differentiation presents a promising therapeutic avenue for mitigating renal inflammation and slowing fibrosis, as exemplified by the use of cordyceps protein (WCP), which significantly reduced CD4+ and CD8+ T cell infiltration and suppressed Th1 differentiation in LN [99].

## 8. Treatment of Renal Fibrosis

The treatment of renal fibrosis in LN patients remains a significant challenge. Current therapeutic strategies primarily focus on suppressing inflammation, reducing immune-mediated damage, and managing associated complications. Immunosuppressive agents, such as corticosteroids and cyclophosphamide, are commonly used to control the inflammatory response in LN [1]. Among these, mycophenolate mofetil (MMF) has shown additional antifibrotic effects in preclinical models. In NZBWF1/J mice, MMF reduced mesangial proliferation, extracellular matrix (ECM) expansion, and tubular atrophy, while significantly decreasing TGF-β expression in both glomerular and tubular compartments [31]. In vitro studies using human mesangial cells stimulated with anti-dsDNA antibodies further demonstrated that MMF treatment led to diminished secretion of TGF-β, reinforcing its potential antifibrotic role [31].

### 8.1. Interventions Targeting TGF-β

Despite the appeal of directly targeting TGF-β in fibrosis, clinical trials in diabetic nephropathy (DN), acute kidney injury, and focal segmental glomerulosclerosis (FSGS) have largely been unsuccessful [41]. The complexity of TGF-β signaling, which plays both fibrotic and regenerative roles, presents a major challenge. A potentially more effective approach may involve targeting downstream components of the TGF-β/Smad pathway rather than inhibiting TGF-β itself.

### 8.2. Emerging Antifibrotic Therapies

Given the central role of fibrosis in CKD progression, several antifibrotic agents approved for pulmonary fibrosis have been investigated for renal fibrosis, although none are currently approved for kidney diseases. Pirfenidone, a TGF-β inhibitor and reactive oxygen species (ROS) scavenger, has shown modest benefits in patients with FSGS by slowing CKD progression, although its efficacy in LN fibrosis remains unclear [100]. Nintedanib, a tyrosine kinase inhibitor (TKI) targeting PDGFR, VEGFR, and FGFR, has demonstrated antifibrotic effects in experimental models of obstructive nephropathy [100]. However, concerns about its potential association with renal thrombotic microangiopathy (TMA) have limited its clinical application in kidney disease [100].

Iguratimod, an antirheumatic drug approved for rheumatoid arthritis in Japan and China, has been shown to reduce renal interstitial fibrosis and tubular atrophy in murine LN by inhibiting collagen deposition and TGF-β/Smad signaling [100]. Additionally, it decreased immune complex deposition and inflammatory cell infiltration, suggesting a dual anti-inflammatory and antifibrotic effect [100].

CAR-T cell therapy, originally developed for cancer, has recently emerged as a potential antifibrotic strategy. CAR-T lymphocytes engineered to target fibroblast activation peptide (FAP) have shown promising results in reducing fibrosis in cardiac and hepatic models [100]. In murine models, FAP-specific CAR-T cells successfully eliminated activated fibroblasts, leading to fibrosis regression and improved organ function [100]. While still in the early stages, this strategy could be adapted for renal fibrosis by identifying kidney-specific fibroblast markers, minimizing off-target effects, and assessing long-term efficacy in chronic kidney disease.

Beyond these, renin–angiotensin–aldosterone system (RAAS) inhibitors have demonstrated antifibrotic potential in murine LN models, in which treatment with ACE inhibitors reduced interstitial fibrosis and tubular atrophy [101]. However, human studies validating these findings are limited, and long-term RAAS inhibition poses risks such as hyperkalemia, necessitating cautious clinical application.

### 8.3. Other Interventions

Non-pharmacological interventions, including dietary modifications and lifestyle changes, may complement pharmacological treatments in managing renal fibrosis in LN, although available data are scarce. High dietary salt intake has been conducive to renal fibrosis and exacerbated murine lupus, suggesting that a diet low in salt may help reduce disease progression [102,103]. Additionally, vitamin K3 has been observed to exert anti-fibrotic effects in MRL/lpr mice by decreasing levels of collagen I, CTGF, and α-SMA [104]. However, further research is needed to determine the optimal dietary and therapeutic interventions for managing renal fibrosis in LN patients.

### 8.4. Limitations

Despite advances in antifibrotic drug development, several obstacles hinder clinical translation. LN fibrosis results from a combination of immune-mediated injury, maladaptive repair, and contributions from a multitude of pathogenic cascades, rendering therapeutic targeting challenging. Many antifibrotic agents, including TGF-β inhibitors and tyrosine kinase blockers, carry risks such as immune suppression, vascular toxicity, and thrombotic microangiopathy, which limit their long-term use. Fibrosis is often detected late in disease progression, reducing the window for therapeutic intervention.

Successful pre-clinical studies, namely in murine models, have not always translated to successful clinical trials in patients, possibly due to differences in immune system regulation, renal architecture, and/or fibrotic responses between mice and humans. Furthermore, the cellular and molecular drivers of fibrosis in these models may not fully align with those in human disease, underscoring the need for validation using patient-derived tissues. Yet other factors have plagued such trials in LN, including small sample sizes, the absence of repeat biopsies to assess histological change, and the paucity of reliable biomarkers to monitor clinical or histological response.

In this context, several urinary biomarkers have been proposed as potential non-invasive alternatives to renal biopsy for assessing renal fibrosis, and their clinical utility remains an area of active research [105]. Candidate biomarkers and various extracellular matrix proteins have shown promise in detecting renal injury and fibrosis, but their sensitivity and specificity for distinguishing LN fibrosis from other forms of active inflammation warrant further study.

## 9. Conclusions

Both tubulointerstitial fibrosis and glomerulosclerosis characterize the fibrotic kidney in LN, with a wide spectrum of molecular signatures documented, including hyper-expressed α-SMA, collagen, FSP-1, fibronectin, and vimentin. Both resident renal cells—such as TECs, fibroblasts, podocytes, mesangial cells, and parietal epithelial cells—as well as infiltrating macrophages, among others, emerge as key cellular players driving fibrosis, triggered by a multitude of pro-fibrotic effectors, most notably TGF-β. Compared to other CKDs, the cast of cellular and molecular players driving fibrosis have been sparsely investigated in LN. Ongoing spatial transcriptomic and spatial proteomic studies of LN across several laboratories are likely to shed light on this important manifestation of lupus in a comprehensive way. Both advanced imaging technologies and more predictive biomarkers are anticipated to augment our understanding and management of LN fibrosis in the coming years. A better understanding of the cellular and molecular processes driving fibrosis in LN may help inform the development of therapeutic strategies for this disease, as well as the identification of individuals at higher risk of developing ESKD.

## Figures and Tables

**Figure 1 ijms-26-02621-f001:**
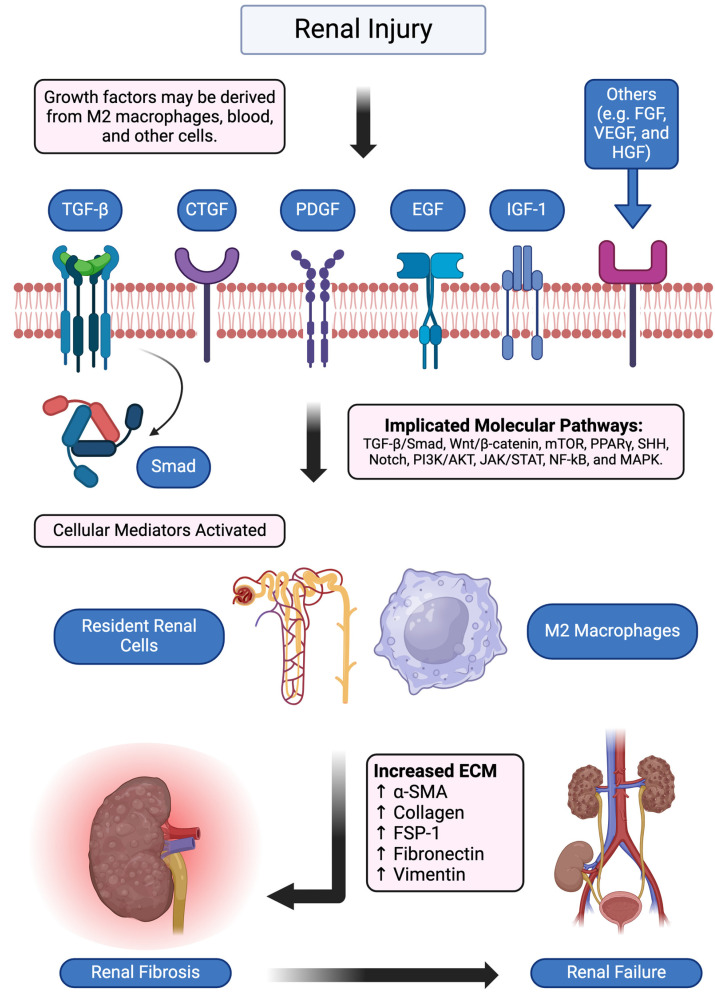
Renal fibrosis roadmap. Stimulated by injury, several molecular mechanisms, including activation of growth factors, signaling pathways, inflammation, and resident renal cells, have been implicated in renal fibrosis. Figure created with BioRender.com. Abbreviations: TGF-β, transforming growth factor-beta; CTGF, connective tissue growth factor; PDGF, platelet-derived growth factor; EGF, epidermal growth factor; IGF, insulin-like growth factor; FGF, fibroblast growth factor; VEGF, vascular endothelial growth factor; HGF, hepatocyte growth factor; α-SMA, alpha-smooth muscle actin; FSP-1, fibroblast-specific protein-1.

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
