# Peer review of "Molecular and Cellular Mediators of Renal Fibrosis in Lupus Nephritis"

_ijms, 2025, doi:10.3390/ijms26062621_

Round 1
Reviewer 1 Report
Comments and Suggestions for Authors
This manuscript presents an insightful analysis of renal fibrosis in patients with lupus nephritis (LN). It is a comprehensive, well-structured, and up-to-date study, incorporating robust data to elucidate and thoroughly investigate this important topic.
A few recommendations for further enhancement:
- Consider including a brief discussion on potential genetic factors that may influence individual susceptibility to renal fibrosis.
- Table 1 – The listed abbreviations lack clarity and would benefit from further specification.
- Line 220 – Ensure that all newly introduced abbreviations are clearly defined upon first usage.
Author Response
Comment 1: Consider including a brief discussion on potential genetic factors that may influence individual susceptibility to renal fibrosis.
Response: Thank you for this important feedback. Available literature surrounding the influence of genetic factors on LN fibrosis is scarce; however, we have added a section titled “Race, Ethnicity, and Genetic Factors” that briefly discusses possible genetic associations. Please see section #4 (Line #359).
Comment 2: Table 1 – The listed abbreviations lack clarity and would benefit from further specification.
Response: Thank you for your suggestion. We have modified the abbreviations in Table 1 to offer more clarity.
Comment 3: Line 220 – Ensure that all newly introduced abbreviations are clearly defined upon first usage.
Response: We thank the reviewer for this suggestion. The abbreviations have been expanded.
Reviewer 2 Report
Comments and Suggestions for Authors
Summary and general comments:
In the manuscript “Molecular and Cellular Mediators of Renal Fibrosis in Lupus Nephritis” the authors performed a literature search followed by a narrative review describing cellular and molecular drivers for kidney fibrosis in patients with lupus nephritis (LN).
Overall, the paper is well organized and excellent, and I really enjoyed reading it, however it needs a little rectification.
Specific comments:
1) Table 1: Abbreviation explanations for “A” and “SMA” are missing. Please add them and make table 1 annotations clearer.
2) Rationale: The rationale was well described with recent references.
3) Material & Methods: All participants and methods were well detailed, allowing easy replication. The statistical analysis is correct.
4) Results: The results are well developed, clear and concise, and I felt it easy to follow authors analyses.
5) Discussion: The authors discussed all key results taking into account the literature previous data
6) Study limitations: The authors also well discussed the limitations of their study
7) English: the English in the manuscript is excellent.
Thanks for the opportunity to review this manuscript. If appropriately corrected, this narrative review could provide valuable information on biomarkers of kidney fibrosis in LN.
Author Response
Comment 1: Abbreviation explanations for “A” and “SMA” are missing. Please add them and make table 1 annotations clearer.
Response: We greatly appreciate the feedback. Table 1 legend has been revised to offer more clarity.
Reviewer 3 Report
Comments and Suggestions for Authors
good work to publish with few comments:
1- This is a good work to publish. It needs some data to add. I am asking why you did not discuss the role of PPAR gamma in renal fibrosis or the mTOR pathway as a potential target of renal fibrosis.
2- You stated that stranded DNA was associated with lupus nephritis and renal fibrosis. Other research data indicated that renal scarring has been correlated with African Americans. Please add more data about autoimmunity and ethnic correlations.
3- Also, I am asking more about the B cells in lupus nephritis or fibrosis.
4- You can add the role of mitochondria in renal fibrosis
See this reference
Zhang Y, Wen P, Luo J, Ding H, Cao H, He W, Zen K, Zhou Y, Yang J, Jiang L. Sirtuin 3 regulates mitochondrial protein acetylation and metabolism in tubular epithelial cells during renal fibrosis. Cell Death Dis. 2021 Sep 13;12(9):847. doi: 10.1038/s41419-021-04134-4. PMID: 34518519; PMCID: PMC8437958.
5- You can add data about long noncoding RNA in fibrosis
6- Discuss the role of T cell in lupus fibrosis
Liao Z, Yang X, He L, Bai J, Zhou X, Yang J, Niu S, Liu S, Guo J. Cordyceps protein alleviates renal injury by inhibiting T cell infiltration and Th1 cell differentiation in lupus nephritis mice. Int Immunopharmacol. 2024 Sep 10;138:112566. doi: 10.1016/j.intimp.2024.112566. Epub 2024 Jun 28. PMID: 38943968.
Author Response
Comment 1: I am asking why you did not discuss the role of PPAR gamma in renal fibrosis or the mTOR pathway as a potential target of renal fibrosis.
Response: We thank the reviewer for this insightful suggestion. In the section titled “Triggers and Signaling Pathways Mediating Renal Fibrosis,” we have added a subsection called “Emerging Signaling Pathways: mTOR and PPARγ” to discuss their role in renal fibrosis. Please see line #240.
Comment 2: You stated that stranded DNA was associated with lupus nephritis and renal fibrosis. Other research data indicated that renal scarring has been correlated with African Americans. Please add more data about autoimmunity and ethnic correlations.
Response: We agree with the reviewer that data about autoimmunity and ethnic correlations enriches the manuscript discussion. In the section titled “Other Pathogenic Drivers of Renal Fibrosis,” we have added a subsection called “Race, Ethnicity, and Genetic Factors” to explore this connection. Please see line #359.
Comment 3: Also, I am asking more about the B cells in lupus nephritis or fibrosis.
Response: We thank the reviewer for this suggestion. We have added a section to the manuscript titled “lymphocytes,” where we evaluate the role of B cells in LN fibrosis. Please see lines #610-624.
Comment 4: You can add the role of mitochondria in renal fibrosis. See this reference
Zhang Y, Wen P, Luo J, Ding H, Cao H, He W, Zen K, Zhou Y, Yang J, Jiang L. Sirtuin 3 regulates mitochondrial protein acetylation and metabolism in tubular epithelial cells during renal fibrosis. Cell Death Dis. 2021 Sep 13;12(9):847. doi: 10.1038/s41419-021-04134-4. PMID: 34518519; PMCID: PMC8437958.
Response: Thank you for the suggestion. In the section titled “Other Pathogenic Drivers of Renal Fibrosis,” we have included a new subsection called “Mitochondrial dysfunction” and incorporated the reference to further enrich the discussion. Please see line #340.
Comment 5: You can add data about long noncoding RNA in fibrosis
Response: Thank you for the suggestion. In the section titled “Other Pathogenic Drivers of Renal Fibrosis,” we have added to the previous subsection on MicroRNAs and included data on the influence of long noncoding RNAs on LN fibrosis. Please see lines #327-338.
Comment 6: Discuss the role of T cell in lupus fibrosis
Liao Z, Yang X, He L, Bai J, Zhou X, Yang J, Niu S, Liu S, Guo J. Cordyceps protein alleviates renal injury by inhibiting T cell infiltration and Th1 cell differentiation in lupus nephritis mice. Int Immunopharmacol. 2024 Sep 10;138:112566. doi: 10.1016/j.intimp.2024.112566. Epub 2024 Jun 28. PMID: 38943968.
Response: We appreciate this excellent suggestion. We have added a new section titled “lymphocytes” to discuss the role of T cells in LN fibrosis while incorporating the reference mentioned. Please see lines #625-635.
Reviewer 4 Report
Comments and Suggestions for Authors
The manuscript provides a comprehensive review of renal fibrosis in lupus nephritis (LN). It highlights the role of podocytes, mesangial cells, fibroblasts, and macrophages in renal fibrogenesis and identifies critical gaps in understanding the molecular basis of fibrosis. The study underscores the urgent need for improved biomarkers and targeted interventions to attenuate renal fibrosis and prevent disease progression.
Comments:
- The relative contribution of different fibrotic signaling pathways (TGF-β, Smad, Wnt/β-catenin, Notch, and PI3K/AKT) is insufficiently delineated. Please provide a clearer comparison of how these pathways interact and which ones are predominant in LN fibrosis.
- The manuscript suggests multiple sources of renal myofibroblasts, but lacks definitive clarification on their primary origin in LN. Offer stronger evidence to distinguish the primary cellular source of renal myofibroblasts in the LN (e.g., resident fibroblasts, epithelial-mesenchymal transition, pericytes).
- Although several biomarkers (α-SMA, collagen, fibronectin, etc.) have been proposed, their comparative sensitivity and specificity for the diagnosis of LN fibrosis is not well discussed. The authors should include a comparative analysis of these biomarkers in terms of diagnostic reliability, specificity, and potential for clinical use.
- The precise molecular factors leading to podocyte depletion in LN remain unclear. Investigate potential mechanisms responsible for podocyte loss, such as immune-mediated damage, oxidative stress, and epigenetic modifications.
- The role of parietal epithelial cells (PECs) in LN fibrosis is mentioned but not fully elucidated in terms of activation. The activation of PECs, including their interaction with podocyte damage and the Notch signaling pathway, needs further discussion.
- The transition between M1 and M2 macrophages in LN fibrosis is mentioned, but the regulatory mechanisms are not well defined. The cytokine networks and transcription factors involved in the polarization of macrophages from M1 to M2 and their contribution to fibrosis need to be further studied.
- The link between hypoxia-induced renal fibrosis and metabolic reprogramming in LN needs further evidence. Integration of recent findings on how hypoxia-inducible factors (HIFs) influence fibrosis progression and their potential as therapeutic targets could improve the value of the manuscript.
- The manuscript relies heavily on mouse models, but lacks an assessment of their translational validity for human LN fibrosis. Please provide a discussion of the limitations of current murine models of LN and their alignment with human pathophysiology.
- The potential of urinary biomarkers to replace renal biopsy as a tool for monitoring fibrosis is mentioned but is not well documented. More data are needed on the clinical application, sensitivity and reliability of urinary biomarkers in monitoring LN fibrosis.
- Although antifibrotic strategies are suggested, their feasibility and clinical trial results are not sufficiently explored. The discussion of antifibrotic drug development, ongoing clinical trials, and challenges in translating preclinical results into therapy deserve further expansion.
Author Response
Comment 1: The relative contribution of different fibrotic signaling pathways (TGF-β, Smad, Wnt/β-catenin, Notch, and PI3K/AKT) is insufficiently delineated. Please provide a clearer comparison of how these pathways interact and which ones are predominant in LN fibrosis.
Response: We greatly appreciate this feedback. In the section titled “Triggers and Signaling Pathways Mediating Renal Fibrosis,” we have included a subsection called “Comparative Analysis of Signaling Pathways” to indicate the pathways predominant in LN fibrosis. Please see line #250.
Comment 2: The manuscript suggests multiple sources of renal myofibroblasts, but lacks definitive clarification on their primary origin in LN. Offer stronger evidence to distinguish the primary cellular source of renal myofibroblasts in the LN (e.g., resident fibroblasts, epithelial-mesenchymal transition, pericytes).
Response: We thank the reviewer for this insightful suggestion. The origins of renal myofibroblasts remains controversial. There is much debate in the literature surrounding the primary cellular source. As a result, we make a note of the current literature and mention the conflicting evidence in place for the progenitors: bone marrow-derived fibroblasts, tubular epithelial cells, endothelial cells, pericytes, and interstitial resident fibroblasts, particularly in the context of LN. Please see lines #416-419.
Comment 3: Although several biomarkers (α-SMA, collagen, fibronectin, etc.) have been proposed, their comparative sensitivity and specificity for the diagnosis of LN fibrosis is not well discussed. The authors should include a comparative analysis of these biomarkers in terms of diagnostic reliability, specificity, and potential for clinical use.
Response: Thank you for this suggestion. The section titled “Pro-fibrotic Markers of Renal Fibrosis” has been revised to discuss markers of fibrosis, with a newly added subsection called “Comparative Analysis of Fibrosis Markers in LN”. It should be noted, however, the goal of tracking expression of markers such as α-SMA, collagen, fibronectin, etc. is not to establish their diagnostic potential for LN; hence terms such as diagnostic reliability, specificity, sensitivity do not apply. These are simply markers of fibrosis (in any disease). Knowing which cells express which of these markers will shed light on the underlying pathogenic mechanisms, and possibly point to novel therapeutic targets.
Comment 4: The precise molecular factors leading to podocyte depletion in LN remain unclear. Investigate potential mechanisms responsible for podocyte loss, such as immune-mediated damage, oxidative stress, and epigenetic modifications.
Response: We agree with the reviewer that the section could use more clarity. Therefore, we have included a subsection titled “Potential Mechanisms of Podocyte Loss in LN,” where we discuss the immune-mediated damage, oxidative stress, and epigenetic modifications driving this process. Please see line #467.
Comment 5: The role of parietal epithelial cells (PECs) in LN fibrosis is mentioned but not fully elucidated in terms of activation. The activation of PECs, including their interaction with podocyte damage and the Notch signaling pathway, needs further discussion.
Response: We agree with the reviewer that the section would benefit from further discussion. Therefore, we have included a subsection titled “Interdependence of Podocytes and PECs in LN” and “Notch Signaling and PEC Activation in LN” to discuss the mentioned interplay. Please see line #505 and #518.
Comment 6: The transition between M1 and M2 macrophages in LN fibrosis is mentioned, but the regulatory mechanisms are not well defined. The cytokine networks and transcription factors involved in the polarization of macrophages from M1 to M2 and their contribution to fibrosis need to be further studied.
Response: We thank the reviewer for this suggestion. We have added a section titled “Regulatory Mechanisms of Macrophage Polarization in LN Fibrosis” to further elaborate their role. Please see line #584.
Comment 7: The link between hypoxia-induced renal fibrosis and metabolic reprogramming in LN needs further evidence. Integration of recent findings on how hypoxia-inducible factors (HIFs) influence fibrosis progression and their potential as therapeutic targets could improve the value of the manuscript.
Response: We appreciate the reviewer’s feedback. In the subsection titled “Cellular Stress” within the “Other Pathogenic Drivers of Renal Fibrosis” heading, we have included a discussion of HIFs and their influence on fibrosis progression as well as their promise as therapeutic targets. Please see lines #299-319.
Comment 8: The manuscript relies heavily on mouse models, but lacks an assessment of their translational validity for human LN fibrosis. Please provide a discussion of the limitations of current murine models of LN and their alignment with human pathophysiology.
Response: We agree with the reviewer that limitations of murine models must be addressed. Under “Limitations”, we discuss their shortcomings. Please see lines #703-707.
Comment 9: The potential of urinary biomarkers to replace renal biopsy as a tool for monitoring fibrosis is mentioned but is not well documented. More data are needed on the clinical application, sensitivity and reliability of urinary biomarkers in monitoring LN fibrosis.
Response: Thank you for this suggestion. Under “Limitations”, we include a brief discussion on how urinary biomarkers are still an area of active research in LN fibrosis. Please see lines #711-715.
Comment 10: Although antifibrotic strategies are suggested, their feasibility and clinical trial results are not sufficiently explored. The discussion of antifibrotic drug development, ongoing clinical trials, and challenges in translating preclinical results into therapy deserve further expansion.
Response: We greatly appreciate the suggestion. In the sections titled “Treatment of Renal Fibrosis,” and “Limitations”, we discuss these limitations. Please see lines #649-678.
Round 2
Reviewer 4 Report
Comments and Suggestions for Authors
The authors have modified the manuscript according to my suggestions.